# Circulating tumour DNA from the cerebrospinal fluid allows the characterisation and monitoring of medulloblastoma

Laura Escudero [1], Anna Llort [2], Alexandra Arias[1], Ander Diaz-Navarro [3,4], Francisco Martínez-Ricarte[2,5], Carlota Rubio-Perez [1], Regina Mayor[1], Ginevra Caratù[1], Elena Martínez-Sáez[2], Élida Vázquez-Méndez[2], Iván Lesende-Rodríguez[6], Raquel Hladun[2], Luis Gros[2], Santiago Ramón y Cajal[2], Maria A. Poca[2,5], Xose S. Puente [3,4], Juan Sahuquillo [2,5], Soledad Gallego[2,5] & Joan Seoane [1,4,5,7✉]

The molecular characterisation of medulloblastoma, the most common paediatric brain tumour, is crucial for the correct management and treatment of this heterogenous disease. However, insufficient tissue sample, the presence of tumour heterogeneity, or disseminated disease can challenge its diagnosis and monitoring. Here, we report that the cerebrospinal fluid (CSF) circulating tumour DNA (ctDNA) recapitulates the genomic alterations of the tumour and facilitates subgrouping and risk stratification, providing valuable information about diagnosis and prognosis. CSF ctDNA also characterises the intra-tumour genomic heterogeneity identifying small subclones. ctDNA is abundant in the CSF but barely present in plasma and longitudinal analysis of CSF ctDNA allows the study of minimal residual disease, genomic evolution and the characterisation of tumours at recurrence. Ultimately, CSF ctDNA analysis could facilitate the clinical management of medulloblastoma patients and help the design of tailored therapeutic strategies, increasing treatment efficacy while reducing excessive treatment to prevent long-term secondary effects.

[1] Vall d'Hebron Institute of Oncology (VHIO), Vall d'Hebron University Hospital, 08035 Barcelona, Spain. [2] Vall d'Hebron Institut de Recerca (VHIR), Vall d'Hebron University Hospital, 08035 Barcelona, Spain. [3] Dpto. de Bioquímica y Biología Molecular, IUOPA-Universidad de Oviedo, 33006 Oviedo, Spain. [4] CIBERONC, Barcelona, Spain. [5] Universitat Autònoma de Barcelona (UAB), 08193 Cerdanyola del Vallès, Spain. [6] Universidade de A Coruña (UDC), 15006 A Coruña, Spain. [7] Institució Catalana de Recerca i Estudis Avançats (ICREA), 08010 Barcelona, Spain. ✉email: jseoane@vhio.net

Medulloblastoma (MB) is the most prevalent malignant brain tumour in childhood. This embryonal tumour of the central nervous system (CNS) is a complex and evolving heterogeneous disease with a wide range of prognosis being in some cases an extremely aggressive malignancy. MB can be subdivided into four molecular consensus subgroups: WNT, SHH, group 3 and group 4. Further subtypes have been proposed[1–5] and a consensus risk-classification scheme that stratify patients in four groups with a 5-year progression-free survival of 91% (favourable), 81% (standard), 42% (high risk) and 28% (very high risk) has been established[4,6].

Treatment for MB includes tumour resection, chemotherapy and, for most patients, craniospinal radiotherapy. Brain tumour surgical resection is currently the first treatment option and surgical specimens are used for diagnosis and tumour characterisation. However, lack of sufficient sample, tissue artifacts or excessive presence of non-tumour tissue can create uncertainty to accurately characterise and classify tumours.

In addition to the interpatient tumour heterogeneity, intratumour heterogeneity has been described in MB and the efficacy of single biopsies to representatively characterise this complexity can be limited. The fact that spatial heterogeneity has been reported at the level of somatic mutations and copy number alterations (CNAs) exemplifies the limitations of single biopsies[7].

Moreover, MB tumours evolve with time highlighting the need for longitudinal samples to characterise tumours at different time points during the progression of the disease. Despite current efforts, around 30% of patients relapse, locally or through metastatic dissemination to the leptomeninges, which is the major cause of mortality in children[8,9]. The comparison of primary and relapsed tumours has shown that specific molecular alterations can appear in the relapse setting[10,11]. In addition, secondary radiotherapy-induced tumours can emerge in some patients[12–14]. To assess metastatic dissemination, cerebrospinal fluid (CSF) cytology together with brain and spinal MRI are performed and patients are classified according to Chang M-staging system[15]. Unfortunately, cytology exhibits limited sensitivity and sometimes cannot provide accurate information.

Altogether, this indicates that there is an underlying and ongoing need to improve the molecular characterisation and monitoring of MB. A relatively noninvasive method to characterise primary and relapsed tumours and monitor the evolution of the disease is required to assist the clinical management of MB patients. In this regard, liquid biopsies and in particular cell-free circulating tumour DNA (ctDNA) have opened an avenue of research[16]. Tumour cells release DNA fragments that circulate in body fluids, and the analysis of ctDNA can provide information about the molecular profile and clinical evolution of cancer[17]. Our group and others have previously reported the detection of CSF ctDNA in adult brain malignancies[18–23]. The analysis of ctDNA in the CSF better reflects the brain tumour genomic alterations than in plasma, it can be used as a molecular diagnostic tool and the dynamic changes during patient monitoring recapitulate the disease course[18,21–25].

Hydrocephalus is very common in paediatric patients with posterior fossa tumours like MB. In these cases and prior to surgical treatment of the tumour, CSF drainage is required to alleviate the intracranial pressure (ICP)[26,27]. In addition, CSF samples are routinely collected for cytologic analysis. However, despite the access to CSF, the study of CSF ctDNA has not yet been explored in the paediatric setting except for a few studies focusing on paediatric diffuse midline gliomas through the analysis of H3 mutations in the CSF ctDNA by droplet-digital PCR (ddPCR) or targeted Sanger sequencing[28,29]. In the current study, we detect ctDNA in the CSF but barely in the plasma of patients with MB. The analysis of CSF ctDNA reflects the genomic alterations of the tumour and allows the subgroup classification and risk stratification of MB patients. In addition, the study of follow-up CSF samples identifies minimal residual disease and reveals the genomic evolution at relapse. Altogether, the analysis of CSF ctDNA is a relatively noninvasive and reliable tool that could aid in the diagnosis, prognosis and monitoring of children with MB.

## Results

**Classification and risk stratification of MB patients**. In order to assess whether the analysis of CSF ctDNA could be a tool for the diagnosis and monitoring of MB patients, tumour, blood and CSF samples were collected from 13 patients (Fig. 1a).

Hydrocephalus was detected through MRI in all patients at the time of hospital admission and CSF drainage, through the implantation of a transitory external ventricular drain (EVD), was required to reduce ICP. Aiming to avoid the implantation of a definitive CSF shunt, 12 of the 13 patients underwent endoscopic third ventriculocisternostomy before the implantation of the EVD. During this intervention, the CSF sample was obtained and used for standard of care clinical measurements (cytology) and research purposes. In the following days, the patients underwent surgical resection of the posterior fossa tumour and additional CSF samples were obtained. Before the administration of chemotherapy, three patients required the implantation of an Ommaya reservoir (intraventricular catheter attached to a reservoir implanted under the scalp) allowing for periodical CSF samples to be collected. For the remaining 10 patients, follow-up samples were acquired through a lumbar puncture.

Whole-exome sequencing (WES) of matching tumour/normal DNA was performed and common MB germline and somatic driver mutations, focal copy number variants (CNVs), arm-level and whole chromosome gain/loss were identified (Fig. 1b and Supplementary Table 1). The frequency of somatic mutations in coding regions was calculated resulting in a median 0.36 mutations per Mb [range: 0.08–1.21], similar to previous studies[2,30].

Patients were classified into previously defined MB subgroups and subtypes[4,5], and were assigned a predicted risk stratification (5-year PFS) based on the molecular subgroup and the presence of risk modifiers (amplification of *MYC* and *MYCN*, metastasis at diagnosis, chromosome 13 loss, large cell anaplastic (LCA) histology)[4] (Fig. 1c).

The clinicopathological characteristics, molecular features, MB-subgroup classification and risk stratification of the 13 MB paediatric patients at the time of enrolment (12 at the time of first-diagnostic and 1 at the diagnostic of relapse (MB8)) are shown in Fig. 1d.

**ctDNA is more abundant in CSF than in plasma**. We sought to evaluate whether ctDNA could be detected in the CSF of the 13 MB patients and how it compared to plasma ctDNA and standard of care CSF cytology. To facilitate ctDNA detection, somatic mutations were selected from the WES of matched tumour/normal samples based on variant allele frequency (VAF) and impact. Mutations were validated by ddPCR in the tumour/normal DNA and then the pre-surgery CSF and plasma cell-free DNA (cfDNA) samples were analysed.

No cells were detected in the CSF from the cytologic analysis for any of the cases. However, ctDNA was detected in the CSF samples obtained before the surgical intervention for 76.9% (10/13) of patients. In contrast, ctDNA could not be detected in plasma, except for one patient albeit with a very low VAF (2.2%) (Fig. 2a). Altogether, these results demonstrate that CSF represents a better source for ctDNA than plasma in MB patients

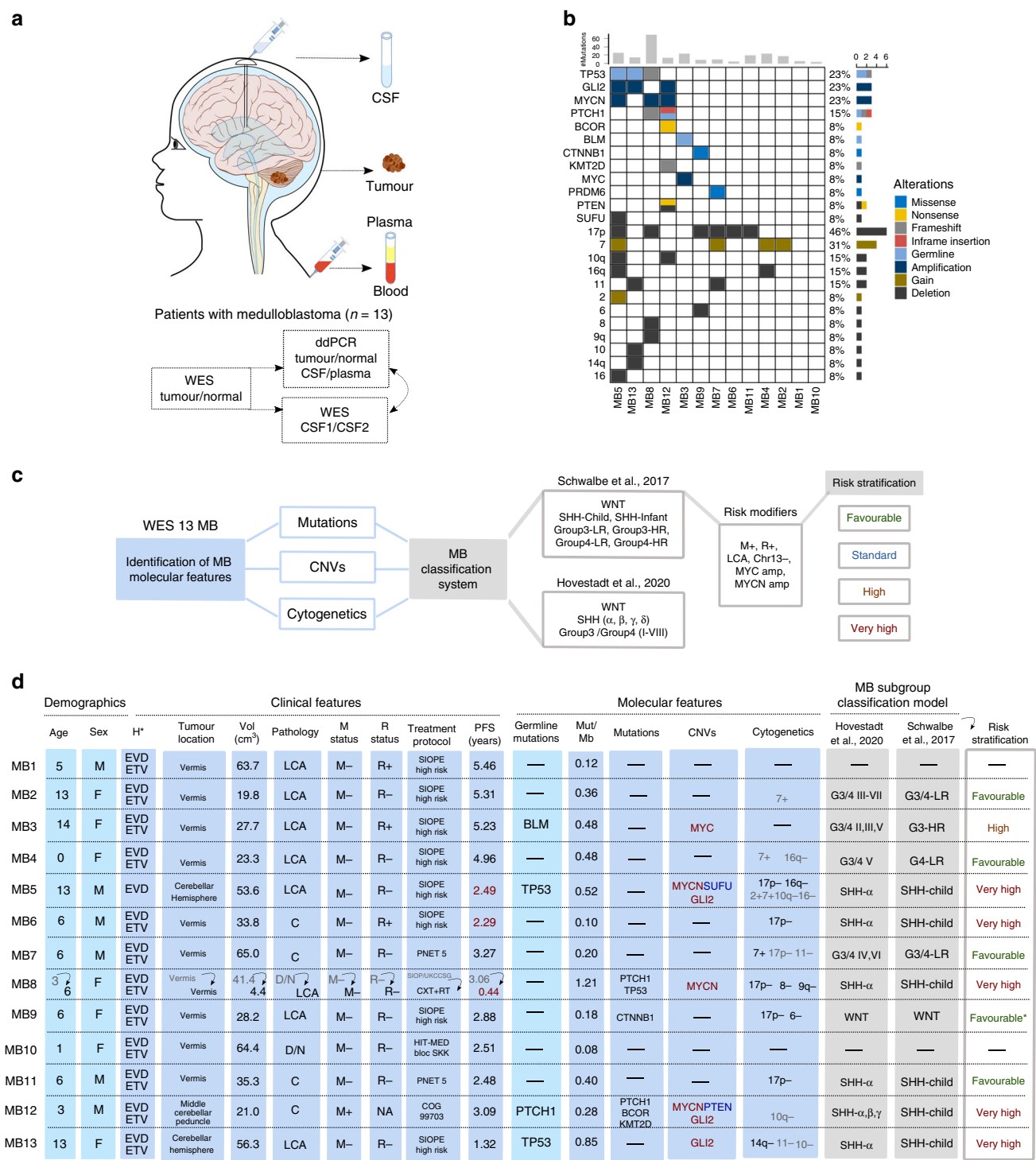

and highlight the higher sensitivity of the ctDNA analysis compared to cytology.

Different levels of CSF ctDNA at diagnosis were observed across the 13 patients. Three of the patients exhibited very high levels (above 25% VAF) when compared to the rest of the patients and were identified as outliers (Fig. 2b). To investigate these differences, the levels of ctDNA (VAF %) and cfDNA (ng/ml) in the CSF were compared with the tumour volume (cm³) but no correlations were observed (Supplementary Fig. 1). Interestingly, the three patients with abundant CSF ctDNA were at very high risk of relapse and disease progression was observed for two of them (MB6 and MB8) at 27.5 and 5.3 months, respectively

(Fig. 1d). These results suggest that high CSF ctDNA levels could be an indicator of more aggressive tumours.

**WES of CSF ctDNA reflects the tumour genomic alterations.** At the time of completing this study, disease progression was not observed for 10/13 patients and CSF ctDNA was not detected during patient monitoring (Supplementary Table 2). In contrast, cancer progression was observed for the remaining three patients (MB5, MB6 and MB8), that were considered at very high risk of relapse (Fig. 1c). A thorough analysis of the tumour/normal DNA and CSF cfDNA at diagnosis and during follow-up or

**Fig. 1 Project outline and characterisation of a cohort of 13 paediatric patients with MB. a** Study schema indicating the patient samples obtained and the methodology used for the analysis. **b** Molecular features, including driver events in MB, identified from the WES characterisation of 13 MB patients. The number of mutations identified for each patient is indicated in the top and the proportion of patients with each alteration on the right. Distinct alterations are indicated in the colour legend. **c** Summary of the MB common molecular alterations investigated for the classification into subgroup and association of risk stratification. **d** Clinicopathological characteristics, molecular features, MB-subgroup classification and risk stratification (based on Schwalbe et al., 2017 5-year progression-free survival (PFS)) for the 13 MB patients. PFS indicated as years until the date of progression (and if not applicable, death) in red or last medical check-up in black. The frequency of somatic mutations in coding regions per Mb (Mut/Mb) was calculated based on a 50.4 Mb (MB1-MB12) and 17.7 Mb (MB13) WES panel. CNV deletion (log$_2$ value < −1, blue), amplification (log$_2$ value >1, red). For the cytogenetic analysis, the threshold for the percentage of arm-level or whole chromosome gain/loss was determined by the mean and the percentiles of the events from the cohort, greater than 75% percentile (>65.5%) in black and greater than the mean (>44.5%) in grey. The risk stratification for patient MB9 was considered Favourable* because WNT subgroup (CTNNB1 & 6-) predominated over SHH (17p-). The definition of the symbols and acronyms are as follows. Hydrocephalus (H): external ventricular drain (EVD), endoscopic third ventriculocisternostomy (ETV). Pathology: large cell anaplastic (LCA), classic (C), desmoplastic/nodular (D/N). Metastasis (M) status: positive (+), negative (−). Residual (R) disease status: >1.5 cm$^2$ (+), <1.5 cm$^2$ (−). Treatment protocol: chemotherapy (CXT), radiotherapy (RT). High risk (HR), Low-risk (LR).

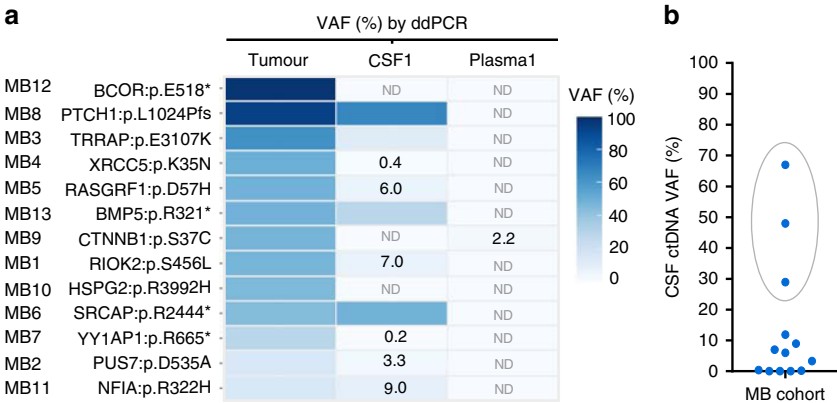

**Fig. 2 ctDNA is abundant in the CSF of paediatric patients with MB. a** Heatmap representing the VAFs analysed by ddPCR from tumour DNA, and CSF and plasma ctDNA samples obtained before surgical resection for the 13 patients. Colour legend for mutant allelic frequencies (VAFs) is shown. VAFs values <10% are indicated. Not detected (ND). **b** Distribution of CSF ctDNA VAF. Outliers identified using ROUT method are highlighted with a grey circle and selected as High CSF ctDNA.

relapse of these three patients, together with another predicted very high-risk patient (MB13), was undertaken.

In order to determine whether WES of ctDNA in the CSF could reliably identify and reflect the genomic alterations of the tumour, we performed WES of the tumour and CSF cfDNA. We found that 98.9% (88/89) of the mutations detected in the primary tumour sample with VAF > 5% could also be detected in the matching CSF ctDNA. Importantly, there was a significant correlation between the tumour VAFs and the ones obtained from the CSF ctDNA (R$^2$ = 0.57, 0.96, 0.53 and 0.87) (Fig. 3a–d). This observation indicates that the CSF ctDNA recapitulated the intratumour heterogeneity present in the primary tumour. Thus, the analysis of CSF ctDNA can provide information about the subclonal genomic architecture of MB tumours.

The analysis of CNVs, arm-level or whole chromosome gain/loss revealed a similar situation. The overall CNV landscape in the CSF samples resembled the one found in the matching tumours (MB6, MB8 and MB13). Despite the restricted amount of cfDNA from MB5-CSF, key MB CNVs detected in the tumour could also be detected in the CSF (Fig. 3e).

**CSF ctDNA facilitates MB-subgroup and risk stratification.** To determine whether the molecular alterations identified from sequencing the CSF ctDNA could provide diagnostic and prognostic information, patients were classified into a MB-subgroup and risk-stratified (Fig. 3e). MB common molecular alterations, including *PTCH1* and *TP53* mutation; *MYCN* and *GLI2* amplification; *SUFU* deletion and 17p loss were detected and facilitated subgroup classification. The four patients were identified as

MB SHH subgroup, for which residual disease, metastatic dissemination, LCA histology or *MYCN* amplification are risk modifiers indicators of very high risk[4]. Importantly, these results showed that the CSF ctDNA analysis could be used as a diagnostic tool to classify and risk-stratify patients.

**CSF ctDNA monitoring identifies minimal residual disease.** We next sought to address whether the WES of CSF ctDNA could assist in the monitoring of the genomic tumour evolution of MB patients and provide information concerning response to treatment or disease progression. Follow-up samples of CSF and plasma were collected in order to determine whether the levels of CSF ctDNA could fluctuate with time during the course of the disease.

Patient MB6 underwent surgical resection and was diagnosed as classic MB. Postoperative imaging identified residual disease. After first-line treatment completion, relapse was observed, and the patient was subsequently treated with second-line chemo- and radiotherapy. Despite an initial reduction in tumour size, sequential MRIs throughout 7 months showed that the leptomeningeal nodule was stable (no change in size was observed) (Fig. 4a). When the follow-up CSF ctDNA was analysed, mutations present in the tumour and in the first CSF were also identified. VAFs of the mutations found in the follow-up CSF showed a substantial decrease as expected due to the decrease in tumour burden caused by the surgical resection (Fig. 4a–c and Supplementary Fig. 2). We were unable to detect additional mutations in follow-up CSF ctDNA; however, we identified 3/5 mutations with a VAF < 1.5% that were

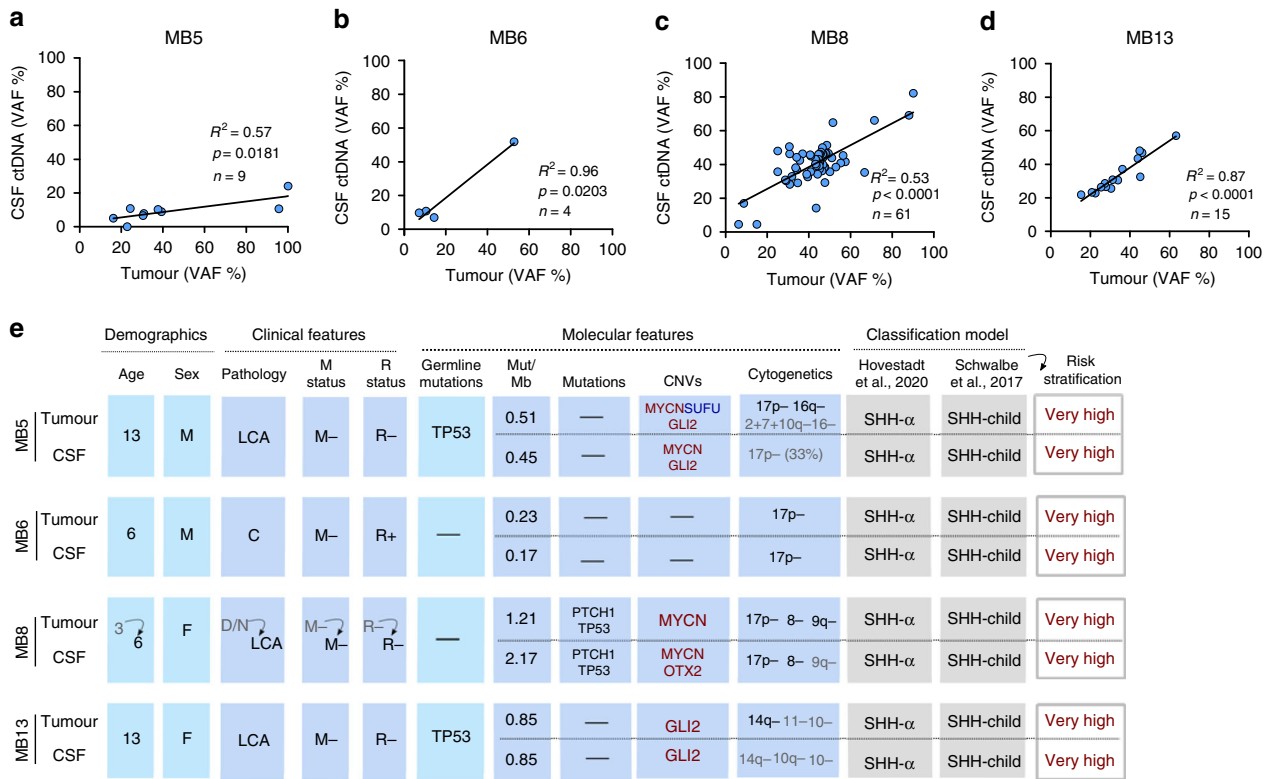

**Fig. 3 WES of CSF ctDNA reveals the genomic alterations of the tumour and identifies MB-subgroup. a–d** Correlation of CSF ctDNA VAF with tumour VAF for each mutation identified in the tumour sample (VAF > 5%). Linear regression and Goodness-of-fit $R^2$ indicated in each figure. n, represents the number of mutations detected in the primary tumour sample with VAF > 5%. **e** Clinicopathological characteristics and molecular alterations based on tumour and CSF ctDNA WES analysis. MB-subgroup classification and risk stratification (based on Schwalbe et al., 2017 5-year PFS). Frequency of somatic mutations in coding regions per Mb (Mut/Mb) based on a 17.7 Mb (MB5, MB6, MB13), 50.4 Mb (MB8-Tumour) and 51.6 Mb (MB8-CSF) WES panel. CNV: deletion (log$_2$ value < −1, blue), amplification (log$_2$ value > 1, red). For the cytogenetic analysis, the threshold for the percentage of arm-level or chromosome gain/loss was determined by the mean and the percentiles of the events from the cohort of 13 MB as indicated in Fig. 1: greater than 75% percentile (>65.5%) in black and greater than the mean (>44.5%) in grey. The definition of the symbols and acronyms are in accordance with Fig. 1 and are as follows. Pathology: large cell anaplastic (LCA), classic (C), desmoplastic/nodular (D/N). Metastasis (M) status: positive (+), negative (−). Residual (R) disease status: >1.5 cm$^2$ (+), <1.5 cm$^2$ (−).

previously detected in both tumour and CSF1 (Fig. 4b). Among these, a stop gain in *SRCAP* (*SRCAP*:p.R2444*) was validated by ddPCR (Fig. 4c and Supplementary Fig. 2). In addition, loss of the chromosome 17p arm was identified in both tumour and CSF1 (Fig. 3e). The fact that a small amount of ctDNA was still observed in the CSF sample obtained at the end of the treatment indicated that the small residual nodule that persisted in the patient could be detected through the analysis of CSF ctDNA (Fig. 4b, c).

A similar case was patient MB13. This patient underwent tumour resection and was diagnosed with LCA MB. Postsurgery MRI determined residual disease <1.5 cm$^2$ while, interestingly, CSF ctDNA was still detected with VAF 7.6% for *BMP5*:p.R321* 13 days after surgery (Fig. 4d, g). Treatment was initiated and no evidence of residual disease was observed during follow-up imaging performed two and a half months after surgery. The mutational landscape and CNV profile of the first CSF collected highly resembled that of the tumour. The analysis of the follow-up CSF samples showed that 3/15 mutations were detected with a VAF < 1% (Fig. 4e, f). Among these, *BMP5*:p.R321* and *MIB1*:p.M468I were validated by ddPCR (Fig. 4g and Supplementary Fig. 2). These results indicated that although no malignancy was observed by MRI, a residual disease was still present since the CSF ctDNA was detected. Both cases, MB6 and MB13, evidenced that CSF ctDNA could facilitate the identification of minimal residual disease in MB patients.

**Analysis of CSF ctDNA reveals genomic evolution.** In contrast to the previous cases outlined, patients MB5 and MB8 progressed and died from their disease following relapse.

Patient MB8 underwent surgery and was diagnosed as a desmoplastic/nodular (D/N) MB. Postoperative imaging identified residual disease <1.5 cm$^2$ and treatment was initiated. Three years postdiagnosis, the patient relapsed and dissemination to CSF was detected by MRI. After a second surgery, the histological appearances changed from D/N MB to LCA MB. Chemo- and radiotherapy treatment was initiated but within 5 months, the patient died (Fig. 5a).

The sequencing analysis of the tumour and CSF samples at relapse revealed common alterations including four predicted drivers: *PTCH1*:p.L1024Pfs, *TP53*:p.N239Tfs, *CDKN2B*:p.G124D and *KMT2A*:p.W1199S (Fig. 5b). Mutations in *PTCH1* and *CDKN2B* were validated using ddPCR in the tumour/normal DNA and the CSF and plasma cfDNA (Supplementary Fig. 2). Importantly, *PTCH1* oncogenic mutations are described as actionable gene mutations for SMO-inhibitors vismodegig and sonidegib (in clinical trials for recurrent or refractory MB showing encouraging results)[31–34]. It is also important to highlight the CNV landscape (Fig. 5c) where the presence of *MYCN* amplification provided information about resistance to SMO inhibitors in MB[35]. In addition to the tumour-specific mutations that were also detected in the ctDNA, CSF-private mutations were identified such as the driver mutation *KMT2A*:p.W1199S that was validated by ddPCR.

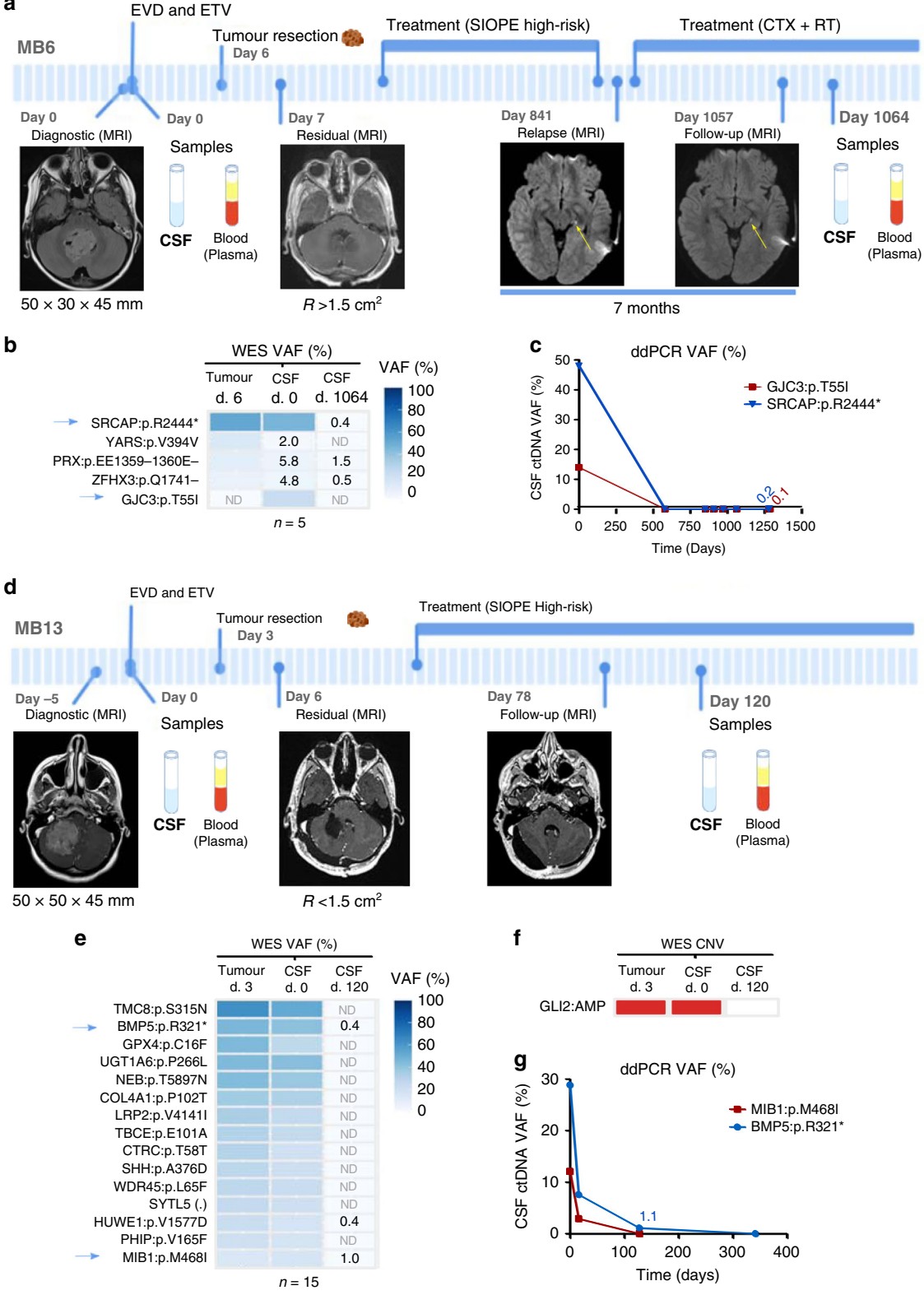

These CSF-specific mutations either revealed intratumour heterogeneity and were missed from the tumour biopsy analysed or indicated the interlesion heterogeneity from the diffused leptomeningeal nodules detected by MRI (Fig. 5a–c and Supplementary Fig. 2).

Similarly, patient MB5 underwent surgical resection and was diagnosed as LCA MB. Postoperative imaging identified residual disease <1.5 cm². Treatment was initiated and during follow-up a

perimedullary leptomeningeal implant at cervical cord was identified. Thirty months after diagnosis, the patient relapsed. Following tumour resection and chemotherapy, an acute tumour progression was observed, and the patient died 9 months after relapse (Fig. 5d).

Despite the limited amount of CSF cfDNA, mutations and most relevant genomic alterations identified in the tumour were also detected in the CSF ctDNA, including *MYCN* and *GLI2*

**Fig. 4 CSF ctDNA characterises the primary tumour and identifies minimal residual disease in patients that respond to treatment.** Longitudinal monitoring of two paediatric patients with MB: **a–c** MB6 and **d–g** MB13. **a, d** Clinical course indicating intervention for hydrocephalus, tumour surgery, treatment protocol, radiologic images (MRI), tumour size or residual disease, and patient samples obtained. Yellow arrow within the MRI indicates the tumour at relapse. **b, e, f** WES analysis of brain tumour DNA and CSF cfDNA before surgery (CSF1) and during follow-up (CSF2). **b, e** Heatmap representing the mutations identified; VAF values <10% were annotated and blue arrows indicate the mutations selected for further validation. **f** Relevant MB CNV identified were indicated in red for amplification ($\log_2 > 1$) or blue for deletion ($\log_2 < -1$). **c, g** Graphical representation of the CSF ctDNA dynamics over time for the selected mutations using ddPCR. Definition of symbols and acronyms: not detected (ND), day (d), external ventricular drain (EVD), endoscopic third ventriculocisternostomy (ETV), residual (R) disease status: >1.5 cm² (+), <1.5 cm² (−), Society of Pediatric Oncology Europe (SIOPE) protocol, chemotherapy (CXT) and radiotherapy (RT).

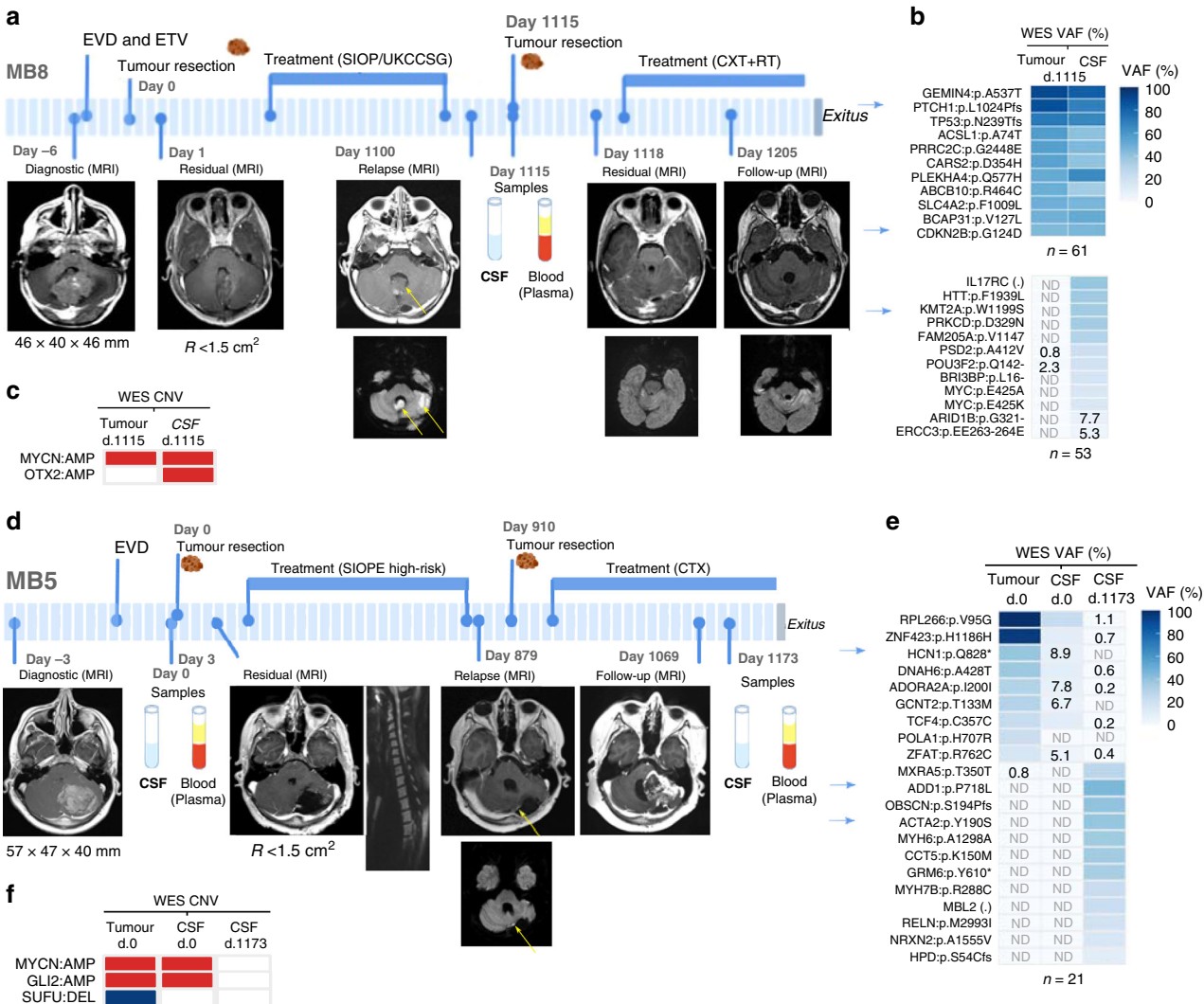

**Fig. 5 CSF ctDNA reveals the evolving genomic landscape at relapse of patients with MB.** Longitudinal monitoring of two paediatric patients with MB that relapsed and died from the disease: **a–c** MB8 and **d–f** MB5. **a, d** Clinical course indicating intervention for hydrocephalus, tumour surgery, treatment protocol, radiologic images (MRI), tumour size or residual disease, and patient samples obtained. Yellow arrows within the MRIs indicate the tumours at relapse. **b, e** Heatmap representing mutations and VAFs (%) identified from the WES of tumour DNA, CSF1 and CSF2 cfDNA samples. VAFs <10% were annotated. For patient MB8, a representative list of mutations (with a high/moderate impact or identified as drivers/predicted drivers by the Cancer Genome Interpreter) is shown. The first heatmap is for mutations detected in both tumour and CSF ($n = 61$) and the second one for private mutations ($n = 53$). Blue arrows indicate the mutations selected for further validation. **c, f** Relevant MB CNV identified were indicated in red for amplification ($\log_2 > 1$) or blue for deletion ($\log_2 < -1$). Definition of symbols and acronyms: not detected (ND), day (d), external ventricular drain (EVD), endoscopic third ventriculocisternostomy (ETV), chemotherapy (CXT) and radiotherapy (RT).

amplification, and partial loss of 17p (Fig. 5e, f and Fig. 3e). Surprisingly, the analysis of CSF ctDNA at relapse revealed a genomic transformation, with a distinct molecular profile not shared with the primary nor with the first relapsed tumour (Fig. 5e and Supplementary Fig. 2). Levels of ctDNA decreased

and were not detected at relapse for the *HCN1*:p.Q828* mutation detected in the primary tumour. In contrast, the levels of ctDNA for alterations identified from the CSF during progressive disease postrelapse were stable or increased throughout 81 days until the patient died (*ADD1*:p.P718L VAF: 41.6-52% and *ACTA2*:p.Y190S

VAF: 13.2-15%) (Supplementary Fig. 2). These results indicate the presence of two independent tumours, one at diagnosis and a different one at relapse. Analysis of germline variants detected by WES revealed that the patient carried a *TP53*:p.R282W mutation, indicating that he suffered from Li-Fraumeni syndrome. Importantly, our data highlights that the analysis of CSF ctDNA can be used to monitor genomic tumour evolution. In this particular case, CSF ctDNA was able to identify the extreme situation of two completely different tumours at diagnosis and relapse.

## Discussion

The diagnosis, the information about prognosis, as well as the identification of therapeutic targets in MB rely on the molecular characterisation of the tumour. To date, in order to characterise MB, tumour specimens are required. This provides limitations since brain tumour surgical procedures may sometimes not be feasible and, even when they are feasible, tumour fragments may provide limited information about the molecular characteristics of the tumour including its intratumour heterogeneity. Moreover, longitudinal biopsies are seldom performed challenging the study of the tumour genomic evolution as well as the characterisation of secondary tumours in the relapse setting. In the present study, we show that the analysis of CSF ctDNA allows the molecular characterisation of MB and facilitates the clinical management of these paediatric patients.

The genomic alterations identified in CSF ctDNA recapitulated those present in the tumour and hence the analysis of the CSF ctDNA constitutes a less invasive alternative to discern the genomic characteristics of MB. Key molecular alterations including, but not limited to, *TP53* and *PTCH1* mutations; *MYCN* and *GLI2* amplifications; *SUFU* deletions and 17p loss were detected in the CSF ctDNA and provided sufficient information for MB molecular diagnosis and risk stratification. Thus, CSF ctDNA allowed the classification of the tumours to the correct subtype and risk subgroup and provided crucial information about prognosis.

Since most MB patients exhibit hydrocephalus[26,27], they require CSF drainage prior to surgery. CSF is then available in a noninvasive way and the CSF ctDNA can be analysed to provide information about the tumour even before surgery. From our cohort, all 13 patients were required to undergo CSF drainage days before surgery. The characterisation of ctDNA in the CSF can thus provide surgeons and oncologists with critical information to evaluate the risk/benefit balances and guide the surgical and chemo- and radiotherapy strategies. Knowing the patient diagnosis and prognosis prior to surgery could help identify those patients with good prognosis that would benefit from a conservative tumour resection and a mild chemo- and radiotherapy dosage. In addition, it would allow the identification of a subset of patients for which residual disease is a risk modifier and an indicator of shorter PFS, highlighting the importance of total tumour resection[4].

As it was shown in other CNS malignancies[18,19,21,24], we observed that CSF is a better source for ctDNA than plasma. Interestingly, we observed that high levels of CSF ctDNA before surgical resection were detected in patients at very high risk of relapse, and disease progression was observed for two of them. Future studies with larger cohorts will be required to investigate whether the abundance of CSF ctDNA could be a valuable prognosis indicator.

Importantly, CSF ctDNA recapitulated the subclonal genomic landscape of the tumour. A significant correlation was observed between the VAFs of the same gene mutations in the tumour and the CSF. This indicates that the study of the CSF ctDNA allows

the characterisation of the intratumour heterogeneity of MB facilitating the identification of small aggressive clones that could promote relapse or resistance to therapy.

During monitoring of the patients through CSF ctDNA, we were able to identify minimal residual disease and assess tumour evolution. The cerebellum is involved in sensory-motor and cognitive functions and MB survivors can suffer from treatment-induced long-term sequelae as well as from the appearance of secondary tumours[36–38]. These patients would benefit from the study of serial CSF ctDNA samples to assess tumour progression and response to treatment, in order to adjust the therapy intensity and duration to the specific status of the tumour at a specific time point. Moreover, CSF ctDNA can complement imaging techniques for early detection of relapse, guiding the therapeutic strategy. Relapse is the major cause of death in MB. Biopsies are rarely performed at relapse and the absence of data about the tumour characteristics may challenge the selection of effective second-line therapeutic strategies. In addition, studies comparing primary versus relapse tumours have shown that other mutations can arise in the recurrent tumour and that secondary tumours can be completely different from the primary tumour[11–14]. In this study, we have shown that WES of CSF ctDNA can characterise the disease at relapse, including the detection of CSF-private mutations that reveal the intratumour/interlesion heterogeneity. In one patient with a Li-Fraumeni syndrome, CSF ctDNA was able to identify the emergence of a completely independent tumour at relapse. In the absence of the CSF ctDNA analysis and since this patient was not suitable for biopsy of the relapsed tumour, the change in the nature of the tumour would be unknown and hence the patient may not have been accurately managed. The analysis of CSF ctDNA at relapse will allow the identification of changes in the presence of therapeutic targets during tumour progression in a way that the treatment strategy could be adapted to the molecular characteristics of the tumour at each time point.

In this proof-of-concept study, we demonstrate that the analysis of CSF ctDNA allows the characterisation of MB and provides an early and accurate molecular diagnosis, including subtyping and risk stratification. It also allows patient monitoring, providing information about minimal residual disease, tumour evolution and the characterisation of the disease at relapse. Studies with larger cohorts are warranted in order to translate our findings into the clinical practice of paediatric patients with MB. This can ultimately lead to tailored therapeutic strategies, reducing unnecessary treatment that would lead to long-term secondary effects and selecting optimal, adjusted, individualised therapies. CSF samples are routinely obtained as part of the standard of care and with the implementation of a CSF-based test in the clinical practice, valuable information could be obtained from each patient, bringing precision medicine closer to MB patients (Fig. 6).

## Methods

**Patient cohort, clinical data and samples**. A total of 13 paediatric patients diagnosed with medulloblastoma were included in the study. All patients were diagnosed and treated at the Vall d'Hebron University Hospital (Barcelona, Spain). A written consent, indicating that the samples obtained as standard of care in the clinical practice could be used for research, was obtained from the parents/legal representant in agreement with the declaration of Helsinki. The study was approved by the Vall d'Hebron University Hospital IRB.

**Sample collection and DNA extraction**. Tumour DNA was extracted from a fresh-frozen or paraffin-embedded section of the tumour biopsy using the QIAamp DNA mini kit (Qiagen) and the QIAamp DNA FFPE tissue kit (Qiagen), respectively. Germline DNA was extracted from peripheral blood cells using the QIAamp DNA mini blood kit (Qiagen). Peripheral blood was collected in K2EDTA-containing tubes (Vacutainer) and plasma was acquired following a $1600 \times g$ centrifugation for 10 min. Both plasma and CSF samples were centrifuged at

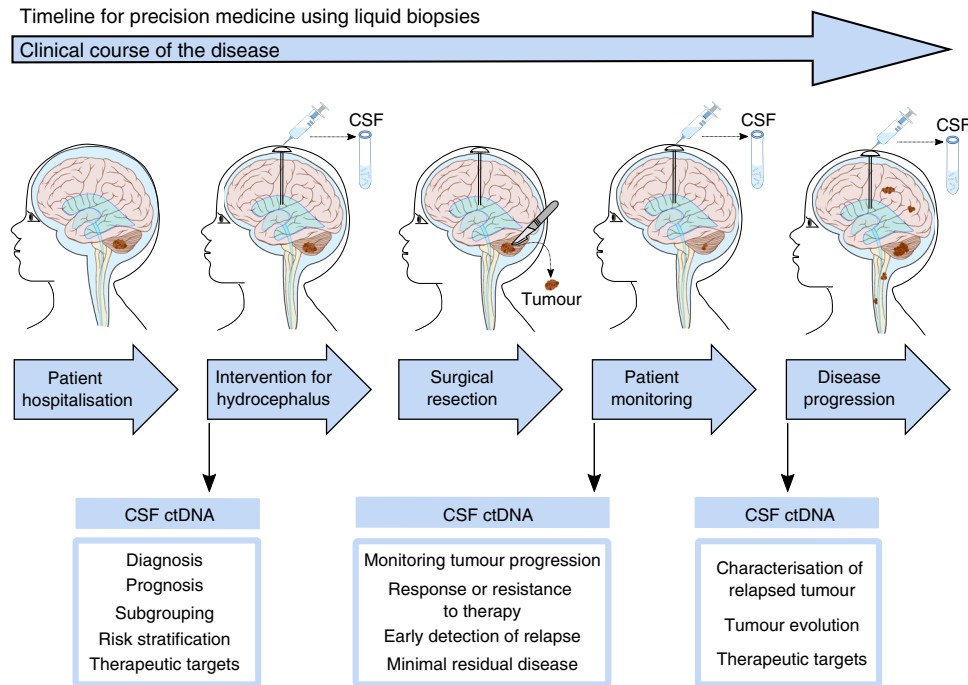

**Fig. 6 Proposed strategy to improve the clinical management of paediatric patients with MB using liquid biopsies.** At the time of patient hospitalisation, MRI allows the detection of hydrocephalus and the brain tumour. CSF drainage is required for patients with hydrocephalus and the CSF sample can be used for WES of ctDNA. The analysis of CSF ctDNA identifies mutations, CNVs, arm-level and chromosome gain/loss and the frequency of somatic mutations. The genomic characterisation of the cancer provides information about MB-subgroup and the risk stratification, that is critical for the diagnosis and prognosis. In addition, therapeutic targets can be identified from the analysis of CSF ctDNA. The information obtained from the CSF ctDNA can guide the surgical resection, identifying those patients for which residual disease is a risk modifier and confers worse prognosis. Analysis of follow-up CSF samples could complement the MRI to identify minimal residual disease or early detection of relapse. Improving the monitoring of tumour progression and identifying response or resistance to therapeutic targets. Importantly, if disease progression is identified, analysis of CSF ctDNA can characterise the relapsed tumour, providing information about the diagnosis, prognosis and potential therapeutic targets. This will be extremely critical for patients with disseminated disease so the therapeutic strategy can be adjusted. Moreover, the comparison of CSF ctDNA from primary with refractory disease would provide information about the emergence of secondary tumours. The study of tumour heterogeneity and cancer evolution would reveal mechanisms of resistance and aid in the discovery of therapeutic targets for MB. Altogether, the analysis of CSF ctDNA can aid in the clinical management of patients and provide information about relapse to advance research towards precision medicine.

$3000 \times g$ for 5 min and the supernatant were collected. cfDNA from plasma and CSF samples were extracted using the QIAamp Circulating Nucleid Acids kit (Qiagen). Genomic and cfDNA were quantified using the Qubit fluorometer.

**DNA sequencing and analysis of genomic alterations**. Whole-exome DNA sequencing (WES) was performed using 2.5 μg of matching tumour and germline genomic DNA samples of the 12 patients. Library preparation and enrichment was performed using SureSelect Human All Exon V5 (Agilent Technologies) and sequenced using Illumina paired-read sequencing platform HiSeq2500 (read length: 2 × 100; coverage: >75× control, >100× tumour; 50.4 Mb). For one patient (MB13), WES was performed using the SureSelect XT HS Human Focused Exome 17.7 Mb as explained below.

Given the limited amount of cfDNA available from the 13 patients, CSF-derived cfDNA from four patients underwent DNA sequencing (6–200 ng). For one patient (MB8), SureSelect Human All Exon V5 (Agilent Technologies) was used for the CSF-derived cfDNA obtained at relapse (51.6 Mb). For the remaining 3 patients, longitudinal CSF samples were obtained before surgical intervention of the tumour biopsy and during progression or follow-up. Matching tumour and germline (200 ng), and follow-up CSF samples (6–200 ng) were analysed using the SureSelect XT HS Human Focused Exome 17.7 Mb (custom constitutional panel, Agilent technologies) for library enrichment and sequenced using Illumina platform NextSeq with a read length of 150 bp.

To process the sequencing data, FASTQ files were aligned to the human reference genome (hg19/GRCh37 hs37d5) using BWA-MEM, and PCR duplicates were removed. Alignment information is indicated in Supplementary Data 1. Somatic mutation calling was performed using Sidrón[39] and annotated using the Variant Effect Predictor (VEP)[40]. Germline mutations were extracted using BCFTOOLS and filtered using common SNPs from dbSNP. Copy Number Variants (CNVs) were detected using the program exome2cnv[41]. Normal samples were compared to each other to identify any region that could be altered only by the region's own variability. A summary with the events was done to obtain the percentage of loss/gain of each chromosome.

Filtering was performed based on the following criteria. For SNVs, we focused on mutations affecting the protein coding sequence; a depth of coverage ≥20 was required to compare the tumour with the ctDNA VAF; if VAF > 5% in any of the samples from the same case, it was considered as a mutation, and its presence in the other samples was studied. To determine whether a sample contained a mutation that was detected in another sample from the same case, a VAF greater or equal to 1% was established as threshold. For germline mutations, we analysed coding regions with a mapping quality ≥50. To choose only heterozygous variants, we focused on 0.35 ≤ VAF ≥ 0.75, and mutations with a VAF greater or equal to 0.001 in the control cohort of the gnomAD database were removed to ensure that polymorphism were not present in the final set of mutations. For CNVs, only regions which included ≥10 probes were considered. Regions with a log2ratio greater or equal to 0.5 were considered gains, and lower or equal to −0.5 were considered deletions. For a more stringent analysis we used −1 ≤ LOG2RATIO ≥ 1 as a threshold to report MB CNVs.

**Classification of MB-subgroup**. To identify mutations, CNVs and cytogenetic features relevant to MB, we focused on the following alterations[4,5]. Mutations: *BCOR, CTDNEP1, CTNNB1, DDX3X, GFI1, GFI1* act, *KBTBD4, KDM6A, KMT2C, KMT2D, PRDM6* act, *PTCH1, SMARCA4, SMO, SUFU, TERT, TERT* promoter, *TP53, TP53* (GERMLINE, GL), *ZMYM3*. CNVs: *GLI2* amp, *MYC* amp, *MYCN* amp, *OTX2* amp, *PTEN* del, *SUFU* del. Cytogenetics: 1q+, 2+, 4−, 5+, 6−, 7+, 8−, 8+, 9p+, 9q−, 10−, 10q−, 11−, 13−, 13+, 14−, 14q−, 15−, 16−, 16q−, 17p−,17p+, 18+, 21−. i17q was excluded because it was not possible to calculate from the data. The number of events from each category were counted and when alterations belonged to distinct subgroups, the group with the most events was selected (Supplementary Table 1).

For the identified germline mutations, those reported as pathogenic and associated with a hereditary cancer predisposing syndrome in ClinVar[42] were annotated (Supplementary Data 2). MB-subgroup information from the germline mutations was based on previous studies[43].

The frequency of somatic mutations per Mb was calculated from the tumour and CSF somatic coding mutations (with VAF > 5%) divided by the WES panel size (Supplementary Data 1, 3–7). The values obtained were compared with the frequency of somatic mutations observed for MB[2,30].

The proportion of arm-level chromosomal gain/loss was calculated for those with log2 value >0.5 and <−0.5; and % chromosome arm loss >15%. For total chromosome gain/loss, both arms needed to be affected. To define a threshold, the proportion of chromosome gain/loss obtained from the 13 tumour samples were plotted ($n = 46$) to determine the quartile distribution. Samples followed a normal distribution (KS normality test), with the following column stats: min: 18%; 25% percentile: 29.5%; median 44.5%; 75% percentile: 65.5%, max: 100%. For the MB classification subgroup, we included the arm-level or whole chromosome gain/loss within the 75% percentile [covered 65.5–100%, black] and above the median [covered 44.5-65.5%, grey].

**ddPCR analysis**. We used droplet-Digital PCR (ddPCR), a robust method with high sensitivity and precision, to determine the presence of ctDNA in the CSF. From the NGS analysis of paired tumour/normal DNA, mutations were selected based on the impact of the mutation, the VAFs and whether they were relevant to MB pathogenesis or identified as driver mutations by the Cancer Genome Interpreter[44]. Custom Taqman SNP genotyping assays for ddPCR were designed for each selected mutation and ddPCR was performed with 10 ng of tumour/normal genomic DNA, and 1–10 ng of CSF and plasma cfDNA, using the QX200 Droplet Digital PCR system (Bio-Rad) according to manufacturer's protocols and the literature[45].

**Reporting summary**. Further information on research design is available in the Nature Research Reporting Summary linked to this article.

## Data availability

The DNA sequencing data have been deposited in the European Genome-phenome Archive (EGA) database under the accession code EGAS00001004651. The data underlying Figs. 1–5 and Supplementary Table 1 are provided as Supplementary Data 1–7. All the other data supporting the findings of this study are available within the article and its supplementary information files and from the corresponding author upon reasonable request. A reporting summary for this article is available as a Supplementary Information file.

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

## Acknowledgements

We would like to thank the patients at the Vall d'Hebron Hospital that were enrolled in the study and their families. The study was undertaken with the support of the Fundación Asociación Española contra el Cáncer (AECC), FERO (EDM), Ramón Areces Foundation, Cellex Foundation, BBVA (CAIMI), the ISCIII, FIS (PI16/01278) and the Juan de la Cierva fellowship (L.E). X.S.P. is supported by Ministerio de Economía y Competitividad (MINECO) SAF2013-45836-R and CIBERONC; A.D.N. is supported by the Department of Education of the Basque Government (grant number PRE_2017_1_0100). We thank CERCA Programme/Generalitat de Catalunya for institutional support.

## Author contributions

L.E. performed the experimental work, analysed and interpreted the experimental and bioinformatics data, co-wrote the manuscript and designed the figures. A.A. and R.M. contributed to collecting, processing and facilitating experimental work. A.L. and S.G. coordinated the clinical aspects of the study and provided clinical samples and information. A.D.N. and X.S.P. carried out the bioinformatics analysis and interpreted the data. G.C. and I.L. contributed towards the WES. C.R.P. contributed to bioinformatics and statistical analyses. E.M.S., S.R.C., E.V., F.M.R., M.A.P, J.S., R.H. and L.G. contributed to the clinical aspects of the study and provided clinical samples. J.S. conceptualised and supervised the study, interpreted research data and co-wrote the manuscript.

## Competing interests

J.S. is a cofounder of Mosaic Biomedicals and has ownership interests from Mosaic Biomedicals and Northern Biologics. J.S. received grant/research support from Mosaic Biomedicals, Northern Biologics, Roche/Glycart and Hoffmann la Roche. X.S.P. is a cofounder and has ownership interest from DREAMgenics. The remaining authors declare no competing interests.
