## [Peer Review File · Nature Communications]

REVIEWER COMMENTS

Reviewer #1 (Remarks to the Author): expert in paediatric medulloblastoma

This is an interesting and potentially important proof of principle paper. Liquid biopsy is highly topical and it will be important to determine whether it is possible to undertake earlier diagnostics, and track disease post treatment using ctDNA derived from the CSF, in the childhood brain tumour medulloblastoma.

The paper is technically sound, well written and clearly presented. The positive aspects of this study are that the team have provided proof of principle that ctDNA can be detected in CSF from medulloblastoma patients and used as a basis for WES, with initial evidence that ctDNA reflects the primary tumour and that its analysis has potential ability to track disease evolution longitudinally, and to detect evidence of tumour heterogeneity. This has not been shown in the disease before in a dedicated medulloblastoma publication, and is thus potentially of value.

Enthusiasm is tempered by the limited number of patients – 13 in total. This is significantly underpowered to assess the clinical significance of any findings or undertake survival analyses. It is not surprising that such relationships cannot be found in the cohort. Such analyses are inappropriate and could be removed.

Importantly, ctDNA is only detectable in 4 of these patients, and the paper thus really just represents descriptive case studies of these 4 patients. This is the main area of novelty, and if it is considered that this is sufficient subject matter to merit publication, should form the focus of the paper.

Ultimately, studies in much larger cohorts will be necessary to assess the whether application of these techniques across many more samples is robust and able to inform the nature, biological relevance and clinical utility of ctDNA sampling in medulloblastoma.

Reviewer #2 (Remarks to the Author): expert in cancer genomics and intratumour heterogeneity

Escudero and colleagues present a genomic analysis of circulating tumour DNA obtained from Cerebrospinal fluid (CSF) from 13 patients with Medulloblastoma. The authors performed whole exome sequencing analysis on the primary tumours and identified considerable levels of ctDNA in CSF samples, but not in plasma. ctDNA samples with high tumour content reflect the genomic composition of the primary tumours very well. The authors find that the highest levels of CSF ctDNA was found among the tumours with the highest molecular risk. The authors follow up on four individuals with repeated cfDNA measurements over the course of treatment, revealing minimal residual disease in two cases, acquired mutations in a late relapse as well as an independent tumour in a patient with undiagnosed Li-Fraumeni syndrome.

The manuscript is very well written, uses clear figures and a sound methodology.

I cannot judge on the practicality of taking CFS samples, which appears to yield high levels of ctDNA enabling a detailed genomic analysis.

The authors' claims are well supported by their data and analysis and the findings are interesting.

My only concern would be the validity of the author's reasoning of the prognostic value of their findings. While it is evident that the highest levels of ctDNA are detected in patients with very high molecular risk, the data appears insufficient to judge whether ctDNA levels provide any additional prognostic value. The logrank test employed by the authors is known to be inaccurate (and often too optimistic) for low sample sizes, see

<https://www.ncbi.nlm.nih.gov/pmc/articles/PMC2950789/>

From a methodological point a permutation approach appears more suited, eg. by the permGS R package.

As this is unlikely to produce a statistically meaningful difference (I'm open to be proven wrong) or improvement over the existing molecular risk stratification, I would suggest that the authors put less emphasis on the prognostic value, but rather stress that tumours in the high molecular risk produce extremely high levels of cfDNA, which is an interesting finding in its own.

On a related note, I'm wondering to which extent cfDNA levels correlate with primary tumour volume.

Reviewer #3 (Remarks to the Author): expert in brain tumour genomics and ctDNA

This manuscript describes the analysis of tumor derived cell free DNA in cerebrospinal fluid from medulloblastoma patients. While the work is well done and represents an important contribution to the brain tumor field, there are a number of gaps in the presentation and interpretation of the data.

Comments:

(1.) The study cohort is very small (n=13 patients) and the majority of the manuscript focuses on the description of individual patients. This data is hard to follow. The authors should include a Table in the main manuscript which lists the age and gender of the patient, tumor location, tumor volume, MB subtype, therapies, and progression-free survival.

(2.) The authors use a combination of ddPCR and WES for the detection of ctDNA. It is not clear to what extent tumor sequencing was required for the interpretation of CSF results. After all, one of the main points of this work is that CSF ctDNA sequencing might (eventually) obviate the need for tumor sequencing.

(3.) Additional cases and a multivariate analysis, considering at least tumor burden and MB subtype, would seem appropriate before suggesting a relationship between CSFctDNA findings and progression-free survival.

(4.) The workflow, presentation of the data, and conclusions are almost identical for what has been reported in adult diffuse glioma (PMID: 30675060). These similarities and differences should be discussed in more detail.

July 28th, 2020

Response letter to the referees

NCOMMS-20-22522-A. "Cerebrospinal fluid circulating tumour DNA allows the characterisation and monitoring of medulloblastoma"

Reviewer #1:

This is an interesting and potentially important proof of principle paper. Liquid biopsy is highly topical and it will be important to determine whether it is possible to undertake earlier diagnostics, and track disease post treatment using ctDNA derived from the CSF, in the childhood brain tumour medulloblastoma.

The paper is technically sound, well written and clearly presented. The positive aspects of this study are that the team have provided proof of principle that ctDNA can be detected in CSF from medulloblastoma patients and used as a basis for WES, with initial evidence that ctDNA reflects the primary tumour and that its analysis has potential ability to track disease evolution longitudinally, and to detect evidence of tumour heterogeneity. This has not been shown in the disease before in a dedicated medulloblastoma publication, and is thus potentially of value.

We thank the reviewer for considering our manuscript interesting and potentially an important proof of principle paper, as well as, technically sound, well written and clearly presented.

Enthusiasm is tempered by the limited number of patients – 13 in total. This is significantly underpowered to assess the clinical significance of any findings or undertake survival analyses. It is not surprising that such relationships cannot be found in the cohort. Such analyses are inappropriate and could be removed.

We thank the reviewer for the comment that has been raised by the other two reviewers. Following all three reviewers' comments, we have removed the survival analysis. The detailed explanation is shown within reviewer #2 section.

Importantly, ctDNA is only detectable in 4 of these patients, and the paper thus really just represents descriptive case studies of these 4 patients. This is the main area of novelty, and if it is considered that this is sufficient subject matter to merit publication, should form the focus of the paper.

Ultimately, studies in much larger cohorts will be necessary to assess the whether application of these techniques across many more samples is robust and able to inform the nature, biological relevance and clinical utility of ctDNA sampling in medulloblastoma.

We agree with the reviewer. In addition to indicating that this is a proof of concept study we have added the following sentence in the discussion *"Studies with larger cohorts are warranted in order to translate our findings into the clinical practice of paediatric patients with MB."*

Reviewer #2

Escudero and colleagues present a genomic analysis of circulating tumour DNA obtained from Cerebrospinal fluid (CSF) from 13 patients with Medulloblastoma. The authors performed whole exome sequencing analysis on the primary tumours and identified considerable levels of ctDNA in CSF samples, but not in plasma. ctDNA samples with high tumour content reflect the genomic composition of the primary tumours very well. The authors find that the highest levels of CSF ctDNA was found among the tumours with the highest molecular risk. The authors follow up on four individuals with repeated cfDNA measurements over the course of treatment, revealing minimal residual disease in two cases, acquired mutations in a late relapse as well as an independent tumour in a patient with undiagnosed Li-Fraumeni syndrome.

The manuscript is very well written, uses clear figures and a sound methodology.

We thank the reviewer for these words.

I cannot judge on the practicality of taking CFS samples, which appears to yield high levels of ctDNA enabling a detailed genomic analysis.

We thank the reviewer for his/her observation and we would like to provide more information on the matter. Due to the location of the tumor in the posterior fossa, most patients develop hydrocephalus and require an intervention to drain the excess CSF. In addition, CSF samples are obtained during follow-up as a clinical routine for standard of care cytology analysis. Therefore, cfDNA can be obtained and analyzed from the intervention of hydrocephalus or during follow-up without requiring an additional lumbar puncture.

The authors' claims are well supported by their data and analysis and the findings are interesting.

My only concern would be the validity of the author's reasoning of the prognostic value of their findings. While it is evident that the highest levels of ctDNA are detected in patients with very high molecular risk, the data appears insufficient to judge whether ctDNA levels provide any additional prognostic value. The logrank test employed by the authors is known to be inaccurate (and often too optimistic) for low sample sizes, see <https://www.ncbi.nlm.nih.gov/pmc/articles/PMC2950789/>

From a methodological point a permutation approach appears more suited, eg. by the permGS R package.

As this is unlikely to produce a statistically meaningful difference (I'm open to be proven wrong) or improvement over the existing molecular risk stratification, I would suggest that the authors put less emphasis on the prognostic value, but rather stress that tumours in the high molecular risk produce extremely high levels of cfDNA, which is an interesting finding in its own.

As the reviewer suggested, we performed the permutation approach permGS R package. Clinical data was updated with the last visit date and we found significantly shorter progression free survival (PFS) for patients with high CSF ctDNA but not for the risk stratification nor for overall survival (OS). The results and methods are indicated below.

However, following the three reviewers' advices, we have decided to remove the survival analysis given the small sample size. We have focused on reporting that the highest levels of CSF ctDNA were identified in three patients with very high-risk of relapse and that disease progression was observed for two of them. The text and figures have been changed.

Analysis:

Kaplan-Meier survival estimator was computed through Python 3.7 *lifelines* library (Davidson-Pilon *et al.* 2020). Progression free survival (PFS) was computed based on the days until the date of progression (and if not applicable, death) or last medical checkup; and overall survival (OS) based on the days until the date of death or last medical checkup.

Kaplan-Meier survival statistics were calculated based on the IPT and IPZ methods (Wang *et al.* 2010), optimized for two-group comparisons of unequal censorship and small sample sizes. IPT and IPZ estimators were calculated with *permGS* R 3.6 package, through *permIPT* and *permIPZ* functions, respectively. Log-rank test statistic was used as weight, the number of permutations set at 10,000 and the survival function used as formula ($Surv(days, event) \sim group$). We considered the maximum p-value of a 50 iteration of IPT and IPZ, respectively.

On a related note, I'm wondering to which extent cfDNA levels correlate with primary tumour volume.

The reviewer raised a very good question. We have calculated the tumor volume (cm³) for all the patients and compared it with the amount of CSF cfDNA (ng/ml) and the CSF ctDNA VAF (%) by ddPCR; however, significant correlations were not identified. The tumor volume of the cases has been added to Fig 1c and the results of the comparisons with cfDNA and ctDNA are shown in Supplementary Fig. 1. The following sentence has been added in the manuscript “*To investigate these differences, the levels of ctDNA (VAF %) and cfDNA (ng/ml) in the CSF were compared with the tumour volume (cm³) but no correlations were observed (Supplementary Fig. 1).*”

Reviewer #3 (Remarks to the Author): expert in brain tumour genomics and ctDNA

This manuscript describes the analysis of tumor derived cell free DNA in cerebrospinal fluid from medulloblastoma patients. While the work is well done and represents an important contribution to the brain tumor field, there are a number of gaps in the presentation and interpretation of the data.

We thank the reviewer for considering that our work is well done and represents an important contribution.

Comments:

(1.) The study cohort is very small (n=13 patients) and the majority of the manuscript focuses on the description of individual patients. This data is hard to follow. The authors should include a Table in the main manuscript which lists the age and gender of the patient, tumor location, tumor volume, MB subtype, therapies, and progression-free survival.

We appreciate the reviewer’s suggestion and, as suggested by the reviewer, the following information has been added to the table of Figure 1c: Age, Sex, MB pathology subtype (Pathology), tumor location, tumor volume, therapies, progression-free survival and MB subgroup.

(2.) The authors use a combination of ddPCR and WES for the detection of ctDNA. It is not clear to what extent tumor sequencing was required for the interpretation of CSF results. After all, one of the main points of this work is that CSF ctDNA sequencing might (eventually) obviate the need for tumor sequencing.

We thank the reviewer for his/her comment. During our studies, we sequenced both the tumor and the CSF to validate the hypothesis that the CSF ctDNA could capture the mutational landscape of the tumor. We have proved that this is the case and hence we propose that in the future we could use CSF ctDNA to characterize the genomic alterations of the tumor without the need to sequence the tumor. In addition, we have shown that if there is access to tumor samples due to tumor surgical exeresis and samples are sequenced, one can design probes for ddPCR to be able to monitor tumor progression and genomic evolution through the analysis of longitudinal samples of CSF.

(3.) Additional cases and a multivariate analysis, considering at least tumor burden and MB subtype, would seem appropriate before suggesting a relationship between CSFctDNA findings and progression-free survival.

We thank the reviewer for the comment that has been raised by the other two reviewers. Following the three reviewers' advice we have proceeded to remove the survival analysis. The detailed explanation is shown within reviewer #2 section.

(4.) The workflow, presentation of the data, and conclusions are almost identical for what has been reported in adult diffuse glioma (PMID: 30675060). These similarities and differences should be discussed in more detail.

Although there may be some similarities between the present manuscript and the 2019 glioma paper mentioned by the reviewer, we would argue that our manuscript can be as similar to that paper as to other papers including our previous paper De Mattos-Arruda et al 2015 also studying glioma. Similar to our findings in De Mattos-Arruda 2015, where we reported that the presence of CSF ctDNA facilitated the characterization, diagnosis and monitoring of diffuse gliomas and brain metastasis, the 2019 mentioned paper also identify common glioma alterations in the CSF. There are several papers dealing with CSF ctDNA, but these are studies focused on different tumor types with different biology and clinical characteristics than medulloblastoma. To our knowledge, there is no other paper about CSF ctDNA in medulloblastoma. Also, as an important detail, while the 2019 mentioned paper uses targeted sequencing in glioma, we are using the more comprehensive WES that allows better studies of tumor evolution in medulloblastoma. In our current study, we show for the first time that the CSF ctDNA can facilitate the characterization and monitoring of medulloblastoma in order to improve the clinical management of medulloblastoma patients.

REVIEWERS' COMMENTS:

Reviewer #1 (Remarks to the Author):

The authors have addressed the concerns raised. Overall the manuscript has not altered much apart from the removal of the survival studies.

Ultimately, the cohort size is small, and the number of patients with measurable ctDNA for study even smaller (n=4).

Reviewer #2 (Remarks to the Author):

I have no further comments.

Reviewer #3 (Remarks to the Author):

The manuscript has been improved by the authors' revisions. A more thoughtful discussion of the emerging role of CSF ctDNA sequencing in other primary brain tumor types, including diffuse glioma (PMID: 30675060), would considerably strengthen the appeal of this otherwise anecdotal case report series for the broader readership of NatCom.

Here we provide a point-by-point response addressing the reviewer's comments.

REVIEWER'S COMMENTS

Reviewer #1 (Remarks to the Author):

The authors have addressed the concerns raised. Overall the manuscript has not altered much apart from the removal of the survival studies.

Ultimately, the cohort size is small, and the number of patients with measurable ctDNA for study even smaller (n=4).

We thank the reviewer for his/her comments and we are glad that the reviewer considers that we have addressed all his/her concerns.

Please note that we have identified measurable ctDNA in the CSF of 10 out of 13 patients (and we have performed WES of CSF ctDNA of 4 patients at diagnosis and during follow-up or at relapse).

Reviewer #2 (Remarks to the Author):

I have no further comments.

We are glad that the reviewer considers that we have addressed all his/her concerns.

Reviewer #3 (Remarks to the Author):

The manuscript has been improved by the authors' revisions. A more thoughtful discussion of the emerging role of CSF ctDNA sequencing in other primary brain tumor types, including diffuse glioma (PMID: 30675060), would considerably strengthen the appeal of this otherwise anecdotal case report series for the broader readership of NatCom.

We are glad that the reviewer considers that the manuscript has been improved by the revisions. Also, we thank the reviewer for his/her comment and we have cited the article that he/she wants to be cited (PMID: 30675060), expanding our previous discussion about papers showing CSF ctDNA in other primary brain tumors.